# Establishing a Salvage Endoscopic Electroporation (SEE) Service for Colorectal Cancer: The King’s Protocol for Clinical Implementation

**DOI:** 10.3390/jcm14238436

**Published:** 2025-11-27

**Authors:** Ademola Adeyeye, Amyn Haji

**Affiliations:** 1King’s College Hospital NHS Foundation Trust, London SE5 9RS, UK; 2Faculty of Life Sciences and Medicine, King’s College London, London WC2R 2LS, UK; 3College of Health Sciences and Medicine, Afe Babalola University, Aye 360001, Nigeria

**Keywords:** endoscopic electroporation, calcium electroporation, electrochemotherapy, salvage therapy, implementation, protocol

## Abstract

**Background:** Endoscopic Electroporation (EE) is an innovative minimally invasive therapy that utilises short electrical pulses combined with intratumoural (IT) calcium or IT/intravenous (IV) chemotherapy to induce tumour cell death in colorectal cancer (CRC). Based on electrochemotherapy protocols developed for the treatment of skin cancers, EE has shown promising results in salvage therapy, local tumour control, and symptom palliation, particularly in patients who are unsuitable for surgery or standard treatments. **Objective:** To establish, for the first time, a comprehensive and standardised protocol for setting up a Salvage Endoscopic Electroporation (SEE) service in CRC clinical practice, covering multidisciplinary patient selection, procedural steps, equipment needs, and follow-up care. **Methods:** Drawing from the European Standard Operating Procedures of Electrochemotherapy (ESOPE) and emerging clinical evidence on EE from King’s College London, we detail infrastructure, treatment delivery, and monitoring for CRC. Key procedural elements, safety considerations, and patient management strategies are outlined. Electroporation pulses were delivered using the Conformité Européenne (CE) approved ePORE^®^ electroporation generator and single-use CE-marked EndoVE^®^ probe (Mirai Medical, Galway, Ireland). **Results:** Tumour assessment involves both clinical evaluation and endoscopic imaging, with radiological correlation. EE treatment has been safely carried out under sedation using specialised endoscopic probes, leading to effective local tumour response, symptomatic relief, and improved quality of life. Follow-up schedules allow for timely assessment of treatment response and enable repeat treatments if needed. **Conclusions:** This novel protocol provides a practical framework for centres aiming to implement SEE services, promoting consistency, safety, and better patient outcomes. Future prospective studies will refine indications and improve integration of this approach into colorectal cancer management pathways.

## 1. Introduction

Electroporation-based therapies have progressively advanced over the past twenty years as a viable supplement or alternative to conventional cancer treatments, primarily through their successful application in managing skin tumours. Electrochemotherapy (ECT), the earliest and most established form, was initially developed to treat skin metastases locally and has since demonstrated consistent effectiveness across various tumour types, including melanoma, squamous cell carcinoma, basal cell carcinoma, and breast cancer recurrences on the chest wall [1,2,3,4,5,6]. This technique employs electric pulses to temporarily increase cell membrane permeability, enabling non-permeant chemotherapeutic agents—mainly bleomycin or cisplatin—to penetrate tumour cells at cytotoxic concentrations [7,8]. The approach has achieved a high therapeutic ratio and is supported by multicentre data indicating objective response rates exceeding 80% and complete response rates between 60 and 70% after a single treatment session [9,10,11,12]. One centre investigating the safety and efficacy of high-frequency electroporation in a similar patient cohort reported comparable response rates, with an overall response of 91.3%, including a 79% complete response rate [13].

Building upon these foundational principles, calcium electroporation (Ca-EP) has emerged as a non-chemotherapeutic electroporation strategy, whereby high intracellular calcium concentrations are used to induce rapid tumour necrosis. The method employs intratumoral (IT) injection of calcium chloride or calcium gluconate, followed by electric pulses, which trigger mitochondrial dysfunction, ATP depletion, and subsequent cell death [14,15]. Calcium electroporation avoids the systemic side effects associated with chemotherapeutic agents and has been associated with favourable safety profiles, low toxicity, and cost-effectiveness [16,17]. It has shown significant promise in various solid tumours, including soft tissue sarcomas, head and neck cancers, and more recently, malignancies of the upper and lower gastrointestinal tract [18,19,20,21,22].

Colorectal cancer (CRC) remains a significant global health burden, ranking third in incidence and second in cancer-related mortality worldwide [23]. Up to 20% of CRC patients present with locally advanced or inoperable disease, experience recurrence, or are unfit for, or decline, conventional treatments such as surgery, chemotherapy, radiotherapy, and immunotherapy [24]. For these individuals—particularly those with symptomatic tumours (bleeding, obstruction, pain, tenesmus, change in bowel habits)—a minimally invasive local therapy can be both palliative and help maintain quality of life as a salvage treatment. Endoscopic Electroporation (EE) addresses this need by adapting electroporation principles to the endoluminal setting, using flexible endoscopes and applying localised electric pulses to bowel cancer with or without intratumoural/systemic injection of agents, as mentioned above. EE, unlike standard endoscopic therapies such as argon plasma coagulation (APC) and lasers, is described as a non-thermal form of ablation, offering prolonged symptom relief with a limited systemic burden [9,14]. Furthermore, endoscopic stenting is generally limited to lesions distal to the splenic flexure and above the rectosigmoid junction, where its primary aim is to alleviate obstructive symptoms. Therefore, the role of EE is increasingly recognised in symptom-focused management of CRC, especially in the era of personalised and organ-preserving strategies. ECT may also have an advantage over Ca-EP in treating bulky or highly vascularised tumours [19].

We recently demonstrated at King’s College Hospital (KCH), London, the first experience in the United Kingdom (UK) on the feasibility and early clinical efficacy of Ca-EP in CRC patients unfit for standard therapies [25]. This resulted in a 100% symptomatic response, a 91.7% reduction in blood transfusion requirements, an 86.7% improvement in quality of life, and an 83.3% tumour response, mainly achieved after a single session. These findings have sparked growing interest in establishing dedicated SEE services in academic and high-volume cancer centres across the UK. However, standardising implementation is essential to ensure patient safety, technical reproducibility, and accurate measurement of meaningful outcomes.

The European Standard Operating Procedures of Electrochemotherapy (ESOPE), first published in 2006 and revised in 2018, have served as the essential framework for implementing electroporation therapies across Europe [1]. These guidelines cover all aspects of service setup, including infrastructure needs, patient selection, treatment delivery, safety monitoring, and data reporting. While the original ESOPE documents focused on electrochemotherapy using bleomycin or cisplatin for cutaneous tumours, their methodology offers a solid blueprint for applying electroporation-based methods in new anatomical and clinical settings.

EE demands a pioneering adaptation of the ESOPE principles for various electroporation strategies tailored to the colorectal cancer setting. To the best of our knowledge, such guidelines do not currently exist. Unlike skin tumours, colorectal lesions require complex navigation of anatomical constraints, meticulous bowel preparation, and integration with existing colorectal cancer pathways and multidisciplinary teams. Furthermore, the risk profile of flexible endoscopy procedures (including perforation and bleeding) demands high procedural standardisation and appropriate institutional preparedness.

This review offers a thorough roadmap for establishing an SEE service, based on our experience implementing the procedure at a UK NHS Trust and the ESOPE 2018 guidance [1]. The approach highlights a step-by-step protocol covering regulatory compliance, equipment procurement, training, clinical governance, and long-term outcome measurement. Our goal is that this guide will help facilitate the safe and effective expansion of EE into clinical practice, support further research and innovation, and ultimately contribute to better palliative care for patients with advanced colorectal cancer.

## 2. Institutional and Regulatory Framework

The successful implementation of a SEE service within any healthcare institution, whether a tertiary referral centre or a district general hospital, requires alignment with institutional, ethical, and regulatory frameworks. Since EE is a new therapeutic approach in the endoscopic treatment of colorectal cancer, preparatory steps must focus on both clinical governance and logistical infrastructure before patient treatment begins. This should take place after a team (see details later) has been assembled to operate the service.

### 2.1. Ethical and Regulatory Approval

Before initiating SEE service, approval from the New Clinical Procedures Committee (NCPC) or its equivalent is necessary. Furthermore, notification to the local Research Ethics Committee (REC) or Institutional Review Board (IRB) is required when the treatment is part of a research activity, in line with standard practice. A business case will typically be needed to demonstrate cost-effectiveness and ensure financial sustainability. This will require support from the hospital’s business managers and finance team.

If the service aims to collect outcome data beyond standard care, registration with the local Clinical Effectiveness or Audit committee of the colorectal and gastroenterology departments is also necessary. In addition to notification, thorough discussion and collaboration with the local Colorectal Multidisciplinary Team (MDT) and the regional Cancer Alliance Networks (CAN) are essential. Sites are encouraged to adhere to Good Clinical Practice (GCP) principles and, where possible, register with national or international electroporation registries (e.g., PIONEER registry) to support broader data harmonisation and safety monitoring.

In some jurisdictions, including the UK and the EU, the electroporation device or generator and associated probes must meet CE marking or UK Conformity Assessed (UKCA) approval, and appropriate licences must be in place for their use in human subjects. The leading delivery system is the Conformité Européenne (CE)-marked and Medical Device Directive (MDD)-certified (2027/MDD) ePORE electroporation generator and single-use CE-marked EndoVE bipolar electrode probe (Mirai Medical, Galway, Ireland), as shown in Figure 1.

While calcium gluconate (Apollo Scientific Limited, Manchester, UK) is widely used and licenced for intravenous use, its application for cancer therapy constitutes an “off-label” indication. This will require notification and approval from the hospital’s pharmacy department. Therefore, patient information sheets and consent forms must clearly reflect this, in line with legal and ethical standards for novel therapies. The use of IT or systemic chemotherapy as part of ECT in the endoscopy unit will also require approval and collaboration with medical oncology and pharmacy services. Deep sedation with protocol is required only in selected cases. While there are instances of Nurse Administered Propofol Sedation (NAPS) or Non-anaesthesiology administered propofol (NAAP) [26,27], the standard of care is propofol administered by Anaesthesiologists.

### 2.2. Standard Operating Procedures (SOPs)

The foundation of clinical quality in any new service is a comprehensive Standard Operating Procedure (SOP). The SOP for EE can be adapted locally from this document, as well as the ESOPE 2018 framework. It should clarify the roles and responsibilities of team members (endoscopists, anaesthetists, nurses, and clinical scientists), as well as the pre-treatment assessment (including imaging, biopsy, and bowel preparation). Other SOPs cover procedural steps (calcium/chemotherapy injections, pulse delivery, device operation), post-procedural monitoring and aftercare protocols, as well as the pathways for adverse event monitoring and documentation. These SOPs must be reviewed by the hospital’s Clinical Governance Committee or an equivalent body and integrated into the wider colorectal cancer MDT workflow.

### 2.3. Institutional Resources and Integration

Implementing an SEE service requires careful integration into existing endoscopy and oncology units. Institutions must evaluate whether their current endoscopy facilities can support electroporation equipment, whether sedation/anaesthesia protocols are adequate for EE procedures, and if emergency surgical backup is readily accessible in case of complications such as perforation or bleeding.

Coordinating a seamless referral and scheduling pathway is critical. This includes designating clinical leads, scheduling MDT referral slots, and ensuring clear communication between departments. Institutions should also provide access to histopathology review of follow-up biopsies and include EE in MDT documentation platforms (e.g., EPIC).

### 2.4. Risk Management and Insurance

As with any interventional procedure, a risk assessment must be completed by the institution before service commencement. Hospitals should verify that EE is covered under their clinical negligence insurance and ensure that consent documentation clearly communicates risks, including but not limited to perforation, pain, bleeding, incomplete response, or the need for repeat procedures.

A designated clinical governance lead should be appointed to oversee adverse event reporting, safety audits, and patient feedback mechanisms. For services embedded within academic settings or research trials, additional oversight from the Research and Development (R&D) office is recommended.

### 2.5. Patient Referral and Clinical Indications for Endoscopic Electroporation

Patients considered for EE should be identified through a formal MDT discussion involving colorectal surgeons, medical oncologists, pathologists, radiologists, therapeutic endoscopists, onco-geriatricians (where applicable), colorectal nurse specialists (CNS), patient advocates, and palliative care specialists. This collaborative approach ensures a thorough evaluation of the disease extent, patient fitness, and overall care goals. Before referral, alternative standard treatment modalities—including surgery, radiotherapy, systemic chemotherapy, and brachytherapy—should have been either exhausted, deemed unsuitable, or declined by the patient.

It is essential to clearly define the therapeutic intent of EE with both the MDT and the patient. While most applications are currently palliative—targeting symptom control (bleeding, pain, or luminal narrowing)—some patients may be eligible for salvage therapy, debulking, primary local control, or downstaging in select cases. The expected benefits of treatment must be balanced against the patient’s overall condition, prognosis, and disease trajectory, particularly in those with advanced, metastatic, or recurrent CRC.

Patients referred for EE often show a wide variety of tumour types within the rectum or distal colon. These include polypoid masses with focal bleeding, ulcerated low rectal tumours, and exophytic, annular constricting, or infiltrative intraluminal lesions that cause pain, partial obstruction, tenesmus, or changes in bowel habit. The diversity of these presentations necessitates thorough endoscopic and radiological assessments to determine accessibility and the feasibility of IT injection and pulse delivery. Endoluminal access to the tumour remains a crucial consideration.

Suitable Indications [19,21,25] for EE Include:Inoperable, advanced, or recurrent left colonic or rectal adenocarcinoma presenting with symptoms such as bleeding, pain, or mucous discharge, particularly when not amenable to surgery, chemotherapy, radiation, or stenting.Progressive tumours with anticipated worsening of symptoms, where localised therapy may delay deterioration or prevent emergency interventions such as de-functioning stomas.Symptomatic lesions (e.g., ulcerated, friable, or bleeding tumours) in patients undergoing systemic therapy, where EE may provide local control and improve quality of life.Patient preference for minimally invasive or non-systemic palliative therapy, especially in those declining further chemotherapy or radiotherapy due to prior toxicity or personal values.Elderly or frail patients with poor performance status (Eastern Cooperative Oncology Group-ECOG ≥ 2), for whom conventional surgery or systemic options carry disproportionate risk.

### 2.6. Contraindications

While EE has shown a favourable safety profile and high procedural feasibility, specific absolute and relative contraindications must be considered before treatment initiation. 

These reflect both the electroporation mechanism and the practical aspects of colorectal endoscopy.

### 2.7. Absolute Contraindications

Implanted metallic colon stents, due to the risk of conducting electric pulses through the bowel.Pregnancy and lactation: As the safety of IT calcium electroporation during pregnancy has not been established, pregnant or lactating patients should not undergo EE. Potentially fertile patients must use reliable contraception before, during, and after treatment to avoid foetal exposure.Perforation risk: Patients with transmural ulceration, extensive tumour necrosis, or full-thickness involvement on imaging may be theoretically at elevated risk for colonic or rectal perforation and should not be considered unless surgical backup is immediately available.Inaccessible tumour location: Lesions located beyond the reach of a flexible sigmoidoscope (proximal to the splenic flexure), or those obscured by strictures or impassable narrowing, are not currently suitable for EE.

### 2.8. Relative Contraindications

Severe proctitis or inflammation in the target area (e.g., active IBD, radiation proctopathy) due to risk of unpredictable tissue response or complications.Uncorrected coagulopathy: While not a strict contraindication, patients with an international normalised ratio (INR) > 2 or platelets < 50,000/mm^3^ require correction or delay of the procedure. Anticoagulation may be managed using standard perioperative bridging protocols [28]. EE has a profile similar to that of low-risk endoscopic procedures, and relevant guidelines from the British Society of Gastroenterology (BSG) for anticoagulants/antiplatelets are applicable [28].Pacemakers and implanted defibrillators that cannot be deactivated for >30 min: These are not contraindications to EE using bipolar electrodes, even in close anatomical proximity. However, defibrillators should be deactivated before pulse delivery and reactivated immediately thereafter to prevent inappropriate discharges or malfunction. A discussion with cardiologists and electrophysiologists on the impact of the Bipolar EE probe on cardiac devices should be sought.Poor performance status (ECOG > 3): Although EE is minimally invasive, its application should be restricted to patients likely to tolerate bowel preparation and sedation safely.Circumferential tumour involvement > 75% or active colorectal obstruction requiring immediate surgery.

From our experience, the ideal case to start with is an endoscopically accessible, low-risk (≤Tumour level 3, no lymph nodes-N0, no metastasis-M0, free of Circumferential Radial Margin, negative Extra Mural Vascular Invasion, and no Pelvic side wall nodes), less than 5 centimetres in size, non-bulky, polypoid, ulcerative, flat, bleeding, or painful rectal cancer in an immune-competent and compliant individual. A summary of the referral pathway is highlighted in Figure 2.

### 2.9. Pre-Treatment Evaluation

A thorough clinical assessment is crucial for determining EE eligibility, tailoring procedural planning, and ensuring patient safety.

### 2.10. Medical History

Determine the main symptom, such as bleeding, pain, tenesmus, change in bowel habit, or obstruction, as well as the indication for the procedure. A history of anaesthetic complications, particularly with deep sedation or general anaesthesia, must be documented. EE is usually performed under conscious sedation. However, low rectal tumours with anal involvement at or below the dentate line require deep intravenous sedation (e.g., propofol), and appropriate anaesthetic support should be confirmed. Prior colorectal surgery may alter bowel anatomy or access and must be considered during procedural planning. Radiation therapy may cause proctitis and narrowing of the rectal vault, which could limit access to low rectal tumours on retroflexion. Anorectal conditions such as fissures, haemorrhoids, neoplastic anal skin nodules, and fistula may complicate access and rectal examination. This may necessitate the use of local anaesthetic lubricants and sometimes deep sedation. Information on the type, date, duration, outcome, and nature of any previous oncologic therapy or endoscopy is also required. Calculation of body surface area is necessary for ECT. Special attention should be given to frailty and onco-geriatric assessment, as many of these patients are elderly and frail with multiple comorbidities and medications. Routine review by an anaesthetist is not required unless the patient is scheduled for TIVA or deep sedation with propofol. Bowel preparation should follow institutional colonoscopy guidelines, typically including rectal enemas upon arrival at the endoscopy suite, along with a 4–6 h fast. Oral laxatives are administered when indicated, especially for lesions proximal to the mid-sigmoid colon.

### 2.11. Laboratory Investigations

Histological confirmation of the tumour is crucial. Other histological details, such as grade and molecular status, are also significant. A full blood count, along with renal and liver function tests, should be carried out within one week of the procedure, especially for patients receiving ECT with bleomycin or cisplatin. A coagulation profile is essential for patients on antiplatelets or anticoagulants, as it provides a baseline for assessing bleeding risk after tumour injection. Serum calcium levels are not necessary, since the risk of systemic hypercalcemia at the recommended doses (less than 10–20 mL of 10% solution) is very low, based on our experience and explained by the phenomenon of electroporation-induced local “vascular lock” [19].

### 2.12. Imaging and Endoscopic Assessment

Pelvic MRI (for rectal cancer) and contrast-enhanced thoraco-abdominal CT scans should be used to determine the size, location, and morphology of the rectal lesion. These should be performed no more than 2–4 weeks before the EE. The depth of invasion, circumferential extent, and relationship to nearby organs (prostate, vagina, bladder) are crucial for planning IT injections and ensuring patient safety. Specifically, the Tumour-Node-Metastasis (TNM) stage, as well as Extramural vascular invasion (EMVI), Pelvic side wall (PSW), and circumferential resection margin (CRM), are essential for identifying CRC at high risk of local and/or systemic failure. Diagnostic flexible sigmoidoscopy is mandatory and should be performed before treatment to assess access, visibility, and bleeding status. Any obstruction or impassable lesion must be noted.

### 2.13. Special Considerations

Patients currently receiving systemic therapy (e.g., chemotherapy, immunotherapy) may still undergo EE. In such cases, a spacing of one week before or one day after systemic treatment is recommended to reduce additive toxicity or confounding effects. A combination of EE with checkpoint inhibitors (e.g., anti-PD-1 or anti-CTLA-4 agents) is being actively explored and may induce synergistic anti-tumour immunity [8]. For patients with prior pelvic radiotherapy, tissue perfusion may be reduced, which can limit calcium distribution or impair healing, especially within the first three months post-irradiation. However, EE has still shown clinical success in this group as part of salvage therapy, albeit with close post-procedure surveillance. Pre-procedure electrocardiogram and echocardiogram for patients with heart disease and/or pacemakers are not mandatory unless required for sedation assessment. Known arrhythmias, however, should be reviewed in consultation with a cardiologist. Review with a cardiologist regarding pacemaker advice for bipolar electrodes is also recommended. Anticoagulants can be taken up to 24 h before the procedure, as the profile is similar to that of low-risk procedures in the BSG guidelines [28]. The same applies to antiplatelets. Blood transfusion is not indicated because the risk of significant bleeding is low to almost non-existent.

### 2.14. Infrastructure and Equipment

The safe and effective implementation of an EE service requires a dedicated infrastructure that integrates endoscopy, anaesthesia, oncology, and surgical services. Although the procedure is minimally invasive, it relies on precise energy delivery systems, specialised equipment, and institutional readiness to manage both technical and clinical contingencies. Drawing upon the ESOPE 2018 framework and the initial UK clinical experience, this section outlines the critical infrastructure and equipment needed to establish and sustain an EE programme.

### 2.15. Clinical Setting

EE procedures should be carried out in a high-specification endoscopy suite (or a minor operating theatre) equipped with standard endoscopic image guidance systems, anaesthetic facilities for possible deep sedation or TIVA, resuscitation equipment, and an oxygen supply that is immediately accessible. It is advised that procedures be scheduled with colorectal surgical backup available on site, particularly during the initial service phase or for high-risk patients.

### 2.16. Core Equipment Requirements

The following list constitutes the core equipment and materials essential for the EE procedure:
*A. Electroporation Device*
A CE-marked or UKCA-approved pulse generator capable of delivering square-wave, bipolar, high-voltage electric pulses in a controlled and reproducible way. The only delivery system in this field is the CE-marked and MDD-certified (2027/MDD) ePORE electroporation generator, along with the single-use CE-marked EndoVE probe (Mirai Medical, Galway, Ireland).
○The EndoVE^®^ platform is a novel, endoscope-compatible device specifically engineered to deliver EE to gastrointestinal mucosal and submucosal tumours. The catheter includes a treatment chamber with two bipolar electrodes, into which tissue is drawn by vacuum, enabling maximum tissue contact. The device is mounted on the endoscope through an “O” ring (Figure 3).○The electrode allows for segmental delivery of electrical stimulation following intratumour calcium instillation via the endoscope (Ca-EP) or systemic chemotherapy injection, in accordance with pharmacy administration guidelines.○A foot pedal or remote switch can be used for controlled pulse delivery.A vacuum suction device able to deliver pressure up to 400–500 mmHg.

We recommend referring to the “manual of instructions” accompanying the devices for further information on use.*B. Endoscopic Probes and Accessories*
A standard flexible endoscope with a diameter measuring 9.5–10.5 mm (e.g., adult gastroscope or paediatric colonoscope) to fit into the EndoVE^®^ probe.Standard injection needle catheter for intra-tumoral injection of calcium gluconate or chemotherapy (Bleomycin/cisplatin).Carbon dioxide (CO_2_) insufflation for endoscopic comfort and reduced risk of barotrauma.
*C. Injection and Drug Preparation Materials*
Calcium gluconate 10% solution (typically 10–20 mls), preservative-free, stored under appropriate conditions and verified before use15,000 IU/m^2^ for IV bleomycin in ECT10 mg of IV chlorpheniramine and 100 mg of IV hydrocortisone as premedication for IV bleomycin20 mL syringe for IT injectionSterile saline for flushing
*D. Monitoring and Support Equipment*
Continuous cardiopulmonary monitoring system for patients under sedationConscious sedation management as per BSG guidelines.

### 2.17. Imaging and Diagnostic Support

Institutions must have access to standard imaging and histopathology services, as previously mentioned. EE is an image-guided procedure.

### 2.18. Safety Systems and Equipment Maintenance

Electroporation generator devices must undergo regular calibration and maintenance following the manufacturer’s guidelines. A risk assessment should be performed before the first use of the device in human procedures, and technical staff should be trained in emergency shutdown procedures. Standard infection control measures must be implemented as per BSG guidelines [29].

### 2.19. Personnel and Operational Readiness

The SEE service should be staffed by a trained multidisciplinary team, including a lead endoscopist skilled in both colorectal therapeutic endoscopy and electroporation principles, as well as a clinical electroporation technician or biomedical engineer experienced with the pulse generator and electrodes. An anaesthesiologist or sedationist provides patient monitoring, while a specialist nurse or operating department practitioner assists with patient preparation, consent, and recovery. Access to an on-call colorectal surgeon during procedures is also advisable. In centres with research capacity, including a data manager or clinical research nurse is recommended for tracking outcomes and participating in registries.

### 2.20. Training and Accreditation

The introduction of EE into clinical practice requires structured training and accreditation for all personnel involved in planning, executing, and following up on the procedure. As a new and highly specialised therapy combining principles of therapeutic endoscopy, tumour-directed drug delivery, and electroporation physics, EE demands a multidisciplinary skill set, strict governance, and proven competence to ensure patient safety and procedural consistency.

### 2.21. Clinical Competency Framework

A competency-based approach is recommended for all operators and supporting personnel, with formal documentation of the completion of training in the following areas:*Clinical and Endoscopic Competence*
EE should be delivered by accredited colorectal surgeons or therapeutic endoscopists. The minimum skill set consists of the following four T’s: tip control, Torque of the endoscopy shaft, Therapeutic endo-injections, and turning the patient when appropriate. For more complex procedures, advanced endoscopists require the following skill set:
○Experienced in diagnostic and interventional flexible sigmoidoscopy or colonoscopy (minimum of lifetime procedures recommended by BSG or other relevant regulatory bodies)○Can safely deliver submucosal or IT injections under endoscopic guidance in complex settings○Trained to manage intra-procedural complications such as bleeding, perforation, or patient instability.Operators must also be familiar with the selection criteria for EE, the principles of tumour accessibility, and contraindications.

### 2.22. Electroporation-Specific Training

Operators must demonstrate understanding of electroporation physics, including pulse parameters, voltage settings, electrode configuration, and tissue conductance. Additionally, competence in device setup and troubleshooting, such as connecting the pulse generator, handling probes, performing calibration checks, and using a foot switch or remote trigger, is required. Familiarity with safety protocols, like avoiding direct pulse delivery near metallic implants or pacemakers, is also required.

We recommend that training be provided by device manufacturers (e.g., Mirai Medical) or local device distributors through certified user training programmes, as well as peer-mentorship or supervised training from experienced centres with published EE experience.

### 2.23. Anaesthetic and Nursing Team Training

Anaesthetic and peri-procedural nursing staff should receive training on sedation protocols and patient monitoring specific to EE, including early recognition of vagal responses, post-electroporation cramps, or procedural anxiety. Preparation and handling of calcium gluconate, chemotherapy agents for ECT, endoscopic injection needles, and electroporation probes are necessary. The team will also need training in immediate post-procedural care and discharge criteria.

### 2.24. Certification and Accreditation

Although no formal international credentialing currently exists for EE, institutions are encouraged to require internal certification or competency sign-off after completion of supervised cases (typically 3) and participate in national or European registries (e.g., PIONEER or ESOPE-compliant training networks) for quality assurance and benchmarking purposes. There is a need to align with relevant bodies for ongoing professional development in electroporation and endoscopic innovation, when these are established. An online platform for training, which involves watching a video demonstration, passing a quiz (with a score of at least 80%), physically observing a case, and reviewing the video recording of the trainee’s first case, can also serve as a form of certification. A link to a video demonstration (Appendix A) can be found here (registration required; https://www.kingslive.co.uk/video-library accessed on 2 September 2025).

### 2.25. Continuing Professional Development (CPD)

As EE technology advances, practitioners should engage in regular multidisciplinary team reviews of outcomes, complications, and technical challenges to ensure optimal care. Consistent audits and morbidity/mortality meetings, where EE outcomes are openly discussed, are essential. Participation in national or international conferences, including presentation of institutional experiences, is also encouraged to facilitate knowledge sharing and protocol improvement.

### 2.26. Simulation and Dry Lab Models

These are not usually required as the learning curve for the procedure is relatively short (typically 3 cases).

### 2.27. Patient Information, Consent, and Preparation

Patients should receive clear, comprehensive counselling in both written and Treatment intent verbal formats regarding—palliative, salvage, debulking, or disease controlProcedure—sedation, IT injection, electrical pulse delivery, and expected duration (~30 min in straightforward cases)Expected outcomes—tumour shrinkage, local control, symptom relief (e.g., pain, bleeding)Risks and side effects—including:
○Transient pain or minor self-limiting bleeding○Ulceration or mucosal necrosis○Risk of perforation (rare)○Post-treatment mucoid discharge or tenesmus○Pulmonary fibrosis with systemic bleomycin in ECT.

The side effects can be categorised into 3 A’s: Ablation-related, Access-related (Endoscopic), and Analgesia/sedation/anaesthesia-related.

On the day of the procedure, IV access is established for sedation and fluid administration if required. Bowel preparation is necessary, as previously discussed in the “pre-treatment evaluation” section. Sedation requirements are determined on an individual basis and are sometimes omitted at the patient’s request. The usual medications administered include fentanyl, with or without midazolam. Antibiotics are not used, and antispasmodic agents (e.g., hyoscine butyl bromide) are employed selectively.

### 2.28. Procedural Technique and Treatment Delivery

Successful EE treatment requires a coordinated and systematic approach involving careful tumour assessment, patient preparation, precise IT injection of calcium (Ca-EP) or chemotherapy (ECT), and delivery of electroporation pulses via specialised endoscopic probes. The following sections outline the step-by-step procedural considerations for effective and reproducible treatment, drawing on clinical experience and ESOPE principles adapted for colorectal cancer. They can be summarised into four A’s: Access, Assessment, Action, and Aftercare (Figure 4).

Access (Endoscopic): Endoscopic access uses standard endoscopy techniques as previously described with a lubricated gastrointestinal endoscope (9.5–10.5 mm diameter). Patients are typically positioned in the left lateral decubitus position, but modifications are made as necessary. Vital signs, including oxygen saturation and heart rate, are continuously monitored, and supplementary oxygen is administered via nasal cannula according to BSG guidelines. Conscious sedation, deep sedation, or TIVA are administered as indicated. Avoid high oxygen concentrations (FiO_2_ > 30%) during sedation and immediately afterwards due to the risk of oxidative toxicity in patients with previous bleomycin exposure. Ensure CO_2_ insufflation is utilised to reduce discomfort and decrease the risk of perforation.

### 2.29. Assessment: Size, Number, Morphology, and Other Features

Before treatment, all visible or accessible rectal or distal colonic tumours should be evaluated endoscopically (and radiologically prior to the procedure) to determine the following seven S’s:Site: Rectal, recto-sigmoid, sigmoid, or descending colon. The distance from the anal verge and involvement of the sphincter complex (for low rectal lesions), as well as accessibility and lumen traversability with the flexible endoscope.Single (or multiple)—Number of lesions: Single or multiple; if multiple, contiguous or discontinuous.Size of each lesion: Maximum diameter (cm) and estimated volume (if spherical or cylindrical) to determine the amount of IT injection.Shape (Morphology): Polypoid, flat, ulcerated, infiltrative, annular constricting, exophytic, bulky, or circumferential.Surface: bleeding, ulcerated, dominant nodule.Surrounding: e.g., Radiation proctitis, IBD, Diverticular disease.Synchronous lesions: e.g., Benign Polyps.

Endoscopic photography or video recording is essential for documenting the baseline, enabling future comparison. The endoscope is then removed, and medications (for Ca-EP and ECT) are prepared. The EndoVe device (for Ca-EP and ECT) is then mounted for use.

### 2.30. Action (A): Drug Preparation and Administration

In this protocol, IT calcium gluconate is used for calcium electroporation (Ca-EP). Calcium gluconate is preferred over calcium chloride due to its superior haemostatic properties, enhanced stability, and reduced tissue electrical impedance. Calcium is injected into the tumour before each application of the pulsed electric fields. This is achieved by inserting the endoscopic injection needle through the working channel of the endoscope and injecting calcium gluconate (10% solution; ≤10–20 mL per session). The injection volume is determined based on tumour size, shape, and vascularity. The aim is to treat as much of the available surface area as possible to ensure an effective response. For Ca-EP, 0.5 mL of calcium per cm^3^ of tumour tissue should be sufficient to ensure effective delivery; however, pulse delivery to the injected area must occur as soon as possible after injection. When access is difficult, additional calcium (i.e., 1–2 mL) may be necessary. For effective ECT delivery, the ESOPE guidelines recommend a volume of IT injection of 0.25 mL per cm^3^ of tumour for lesions larger than 1 cm^3^. Further details for systemic ECT are outlined in the ESOPE guidelines [1]. Injections should be performed carefully to avoid extravasation or deep transmural injections, especially in friable or irradiated tissues.

### 2.31. Action (B): Electroporation Delivery

The ePORE generator and vacuum machine are confirmed to be functioning. The EndoVE^®^ probe is mounted on the tip of the endoscope and advanced to the tumour site. The probe can be positioned in the 6 o’clock region of the scope (the classical method recommended by the manufacturer) or explicitly aligned with the tumour during endoscopic assessment (the alternative method), as shown in Figure 5.

The probe is activated using a high-suction vacuum to maximise contact with the tumour. Pulsed electric fields (with 1 cm penetration depth and covering a 2 cm^3^ surface area) are then delivered. For Ca-EP delivery, a pulse is applied immediately after calcium injection to the injected area. After each pulse, the electrode is detached from the tissue. This injection-and-pulse cycle is repeated until all available surface area has been treated. To ensure maximum coverage, overlapping of pulse placements where appropriate is recommended. Details of ECT protocols can be found in the 2018 ESOPE guidelines. Key procedural parameters to record include tumour location and size, estimated percentage of tumour surface treated, volume of calcium gluconate injected, number of pulses delivered, maximum applied current, and lowest tissue impedance. These will be compared across subsequent treatment sessions. Treatment is discontinued after any of the following three P’s: complete treatment of pathology (or treated surface), relief of patient discomfort, or the passage of time within a 40 min therapeutic window, as per ESOPE guidelines [1].

### 2.32. Action (C): Completion and Withdrawal

The EE probe is disengaged by turning off the vacuum, and the endoscope is carefully withdrawn after photo documentation immediately after treatment. The following effects of electroporation may occur during or immediately after treatment and are collectively known as the King’s triad of endoscopic ablation lesion (EAL), as seen in Figure 6. It is, however, possible that successful ablation may occur in the absence of some or all of these signs. From our experience, Narrow Band Imaging (NBI) is not a reliable means of demonstrating the EAL.

Another important sign is the “O-ring” sign, which appears when the O-ring of the EndoVE probe is visible on the endoscopy screen (Figure 7). This indicates a possible dislodgement of the probe and suggests that therapy should be stopped, the scope removed, and the probe properly secured.

Total treatment time should usually be around 30 min when performed by an experienced practitioner for simple cases (i.e., 5 min for access, 5 min for assessment, and up to 20 min for treatment).

### 2.33. Aftercare (Post-Procedural Care)

Patients are monitored in accordance with standard endoscopy post-procedure protocols. All patients are discharged on the day of treatment following 30 min to 1 h of observation. There should be provisions for overnight observation if clinically indicated, due to the high-risk profile of the patients. However, all EE procedures at KCH have been day cases with no re-admissions, except for patients who were inpatients for managing disease-related symptoms and referred for EE as part of their management plan.**Follow-up, Outcomes Assessment, and Retreatments**

### 2.34. Structured Follow-Up Schedule

Follow-up after EE treatment should be tailored to the individual, based on tumour characteristics, patient comorbidities, and treatment goals (palliative, salvage, or disease control). However, a general standardised schedule can be adopted to maintain consistency and enable early detection of adverse effects or incomplete responses.

Week 1: Initial telephone consultation with a physician or nurse to assess pain, bleeding, discharge, and early mucosal reaction. This can be repeated in week 4 to monitor symptoms and initial quality of life (QoL) using tools like the SF-12 questionnaire [25]. If necessary, a face-to-face clinic appointment can be scheduled in week 4 (or earlier) to address any significant issues.Weeks 8–12 (2–3 months): Perform primary response assessment using flexible sigmoidoscopy and targeted biopsy (if applicable); this is the earliest point to confirm a complete clinical response (cCR). Patients may receive retreatment at this visit if clinically indicated, following the initial assessment.Weeks 16 and 24 (4 and 6 months): Continue tumour surveillance and assess quality of life. If the patient remains asymptomatic with cCR, further surveillance can be determined by MDT consensus. Patients with symptom control and stable disease, confirmed through endoscopy and imaging, may be moved to a 6- to 12-month surveillance schedule.

For large, bulky, or circumferential lesions, multiple sessions are typically required. These are usually scheduled every 6 to 12 weeks, though they may occur earlier depending on the circumstances. Our definition of failed treatment is a lack of response or disease progression after 3 to 4 sessions.

### 2.35. Outcome Assessment

Tumour response is mainly evaluated endoscopically and histologically (when salvage therapy or local control are the goals). Our established endpoints, following the Response Evaluation Criteria in Solid Tumors (RECIST) framework include:Complete Clinical Response (cCR): Absence of visible tumour, ulceration, or bleeding; negative targeted biopsy following magnification chromo-endoscopy.Partial Response (PR): ≥30% reduction in tumour size and symptoms, but residual visible or histologically positive tumour.Stable Disease (SD): No significant change in tumour size or symptoms.Progressive Disease (PD): ≥20% enlargement of the tumour or new lesions post-treatment.

Response is documented with serial endoscopic images. Where feasible, responses should always be correlated with MRI of the pelvis (for rectal tumours) and CT scans of the chest, abdomen, and pelvis, with or without PET-CT scans (for metastatic disease). Serial Carcino-Embryonic Antigen measurements are helpful but not compulsory.

### 2.36. Retreatments

Patients may be offered repeat EE in the following scenarios:Incomplete response or residual tumour after initial treatmentFor symptomatic relief (e.g., bleeding, subacute obstruction)Tumour recurrence within the previously treated fieldNew or metachronous lesions elsewhere in the rectum or distal colon

Retreatment intervals can be flexible. While electroporation with bleomycin recommends a minimum of two to four weeks between sessions due to cumulative toxicity, Ca-EP does not require fixed intervals, as both have a favourable safety profile and no cumulative dose limitation [18,19,25].

For most patients, 6–8 weeks after treatment is the ideal time to decide on retreatment, based on symptom resolution as well as endoscopic and radiological images.

### 2.37. Discharge from the SEE Service

We discharge patients in the following settings (3 D’s): Declined further treatment, Disease progression, and debilitation from medical co-morbidities. A patient can be re-enrolled if additional treatment is indicated in the future.

### 2.38. Patient Access to Information

Patients and caregivers should be provided with:Written information leaflets outlining expectations, self-care, symptom management, and when to seek help.Dedicated hospital website pages with educational videos and Frequently Asked Questions (FAQs) about EE.Contact details for the clinical team and CNS for post-procedural support.

Community or palliative care teams need to understand the expected effects post-EE therapy, which may include mild, self-limiting mucoid discharge or bleeding in the initial two weeks of treatment that mimics progression but is often part of the therapeutic effect.

### 2.39. Integration into Clinical Practice

As with cutaneous electrochemotherapy, EE is emerging as a valuable adjunctive or alternative therapy in colorectal cancer, particularly in patients with:Inoperable CRCRecurrent/residual disease after conventional therapySalvage therapy for patients desiring non-operative managementPalliative care needs with bleeding or obstructing tumours.

The favourable safety profile, minimally invasive approach, and ability for repeatable local tumour control make EE a promising element in reducing the intensity of CRC treatment and providing palliation.

Its integration into multidisciplinary treatment protocols requires ongoing training, robust audit mechanisms, and inclusion in national clinical guidelines. Collaboration between different hospitals is also essential. Future prospective trials and registries will further define its role and refine selection criteria.

## 3. Conclusions

The development and implementation of an SEE service for colorectal cancer require a multidisciplinary approach, meticulous planning, and adherence to standardised protocols to ensure safety and effectiveness. This approach is similar to that used by the Significant Polyp and Early Colorectal Cancer (SPECC) service. In fact, the SEE service is an extension of the SPECC service (led by AH) at KCH, and it is quite fitting that “SPECCS” helped AA (and the rest of us) “SEE” the potential of EP in managing this unique patient group! Drawing from the experience of electrochemotherapy in skin cancers and emerging data on calcium electroporation, EE offers a promising minimally invasive option for local tumour control, symptom relief, and possible down-staging in selected CRC patients. The philosophy of this new service can be summarised in these “King’s eight commandments of SEE,” each starting with the letter “S”: Simple, Safe, Successful, Standardised, Sedation-only, Short-duration, Scalable (can be repeated gradually), and Same-day procedure.

The protocol outlined here covers patient selection, pre-procedural preparation, treatment administration, and follow-up, offering a comprehensive framework to assist centres aiming to establish EE. As clinical experience expands and prospective trials develop, additional refinements will arise, strengthening its role in colorectal cancer care as well as other Gastrointestinal cancer pathways.

## Figures and Tables

**Figure 1 jcm-14-08436-f001:**
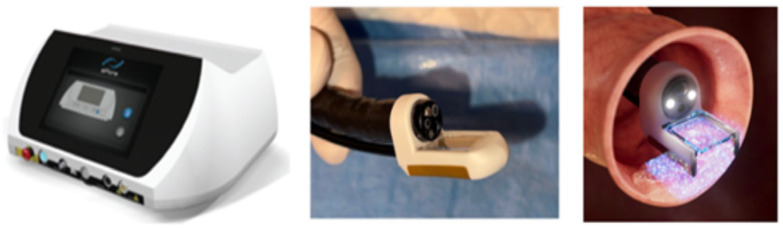
The ePORE^®^ generator (**left**); the EndoVE^®^ electrode (**centre**); and an illustration of the application of pulsed electric fields (electroporation) using EndoVE^®^ (**right**).

**Figure 2 jcm-14-08436-f002:**
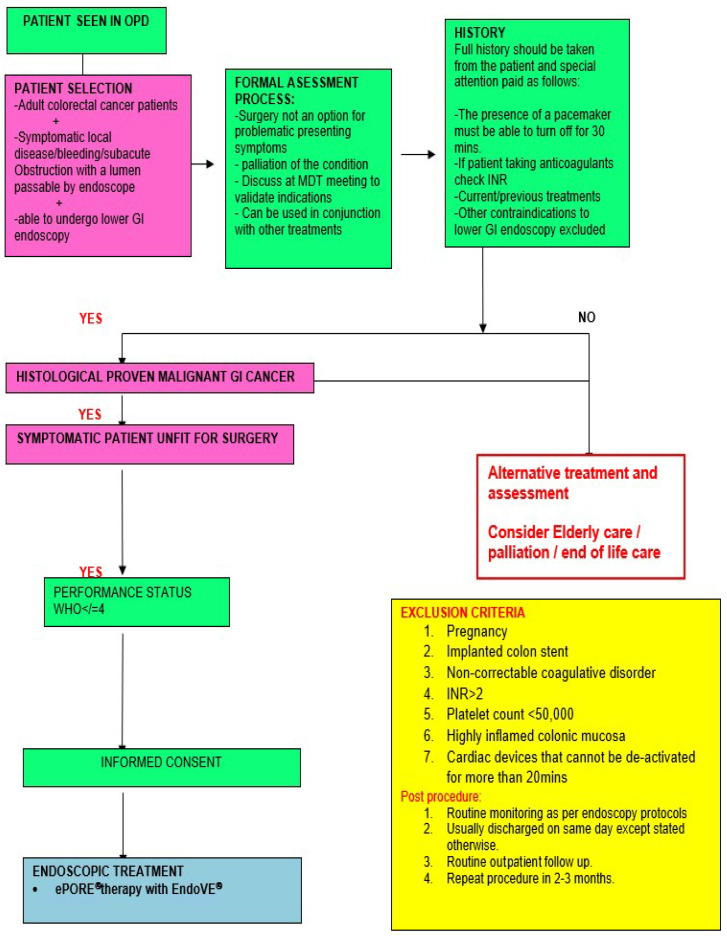
Referral Pathway of Endoscopic Electroporation in Colorectal Cancer.

**Figure 3 jcm-14-08436-f003:**
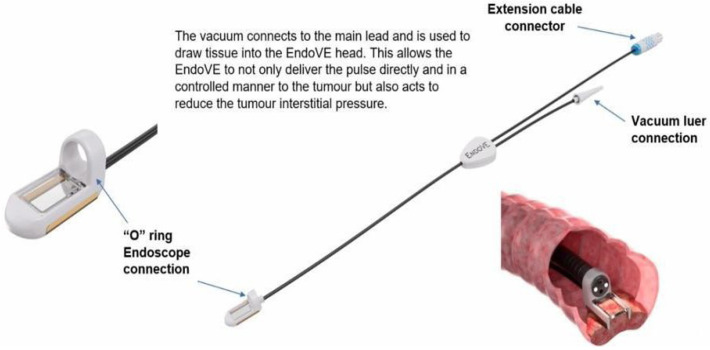
EndoVe connections to the Endoscope (through the “O”-ring), extension cable and Vacuum device.

**Figure 4 jcm-14-08436-f004:**
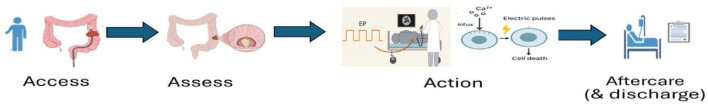
Steps of Endoscopic Electroporation.

**Figure 5 jcm-14-08436-f005:**
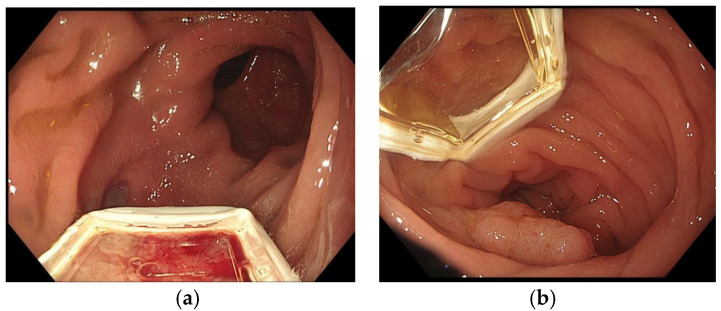
(**a**) Classical attachment of the EndoVe probe corresponding to 6 o’clock position; (**b**) Alternate method of positioning probe outside 6 o’clock (in this case 11 o’clock) position.

**Figure 6 jcm-14-08436-f006:**
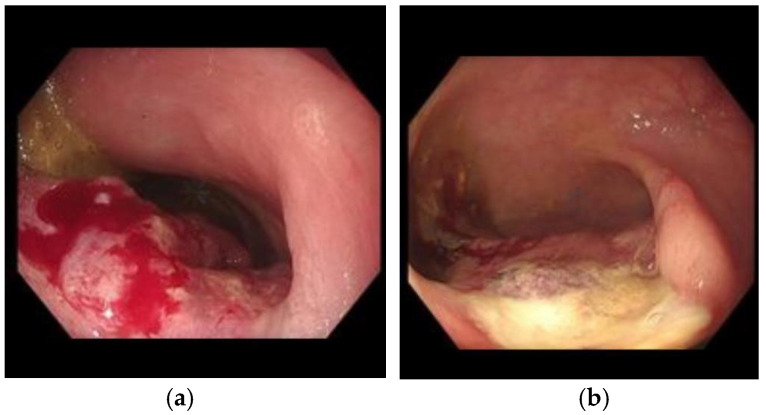
(**a**) Pre-treatment and (**b**) post-treatment images for sigmoid colon cancer treated with EE. The King’s triad of signs suggesting successful EE ablation are seen on the post-treatment image.

**Figure 7 jcm-14-08436-f007:**
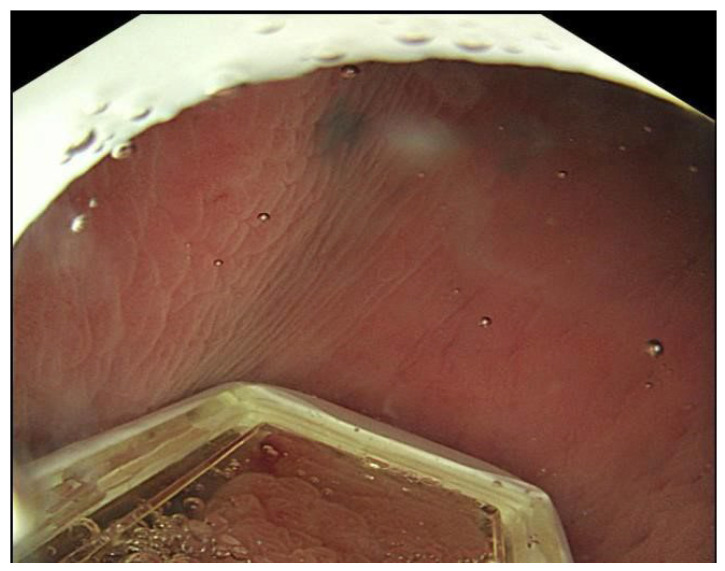
“O”-ring sign (seen in the upper frame of the image) which implies a displaced EndoVE probe.

## Data Availability

No new data were created or analyzed in this study.

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
