# Peer review of "Establishing a Salvage Endoscopic Electroporation (SEE) Service for Colorectal Cancer: The King’s Protocol for Clinical Implementation"

_jcm, 2025, doi:10.3390/jcm14238436_

Round 1
Reviewer 1 Report
Comments and Suggestions for Authors
Authors explain the details of the procedure and pit falls and benefits clearly.
If the authors can elaborate the procedure details in a cartoon form, then it is easier to understand.
Author Response
Reviewer
“The authors explain the details of the procedure, pitfalls and benefits. If the authors can elaborate the procedure details in a cartoon form, then it is easier to understand”.
Response:
We thank the reviewer for their positive evaluation and supportive comments. We agree that a cartoon illustration will make it easier to understand.
Revision Made:
A schematic illustrating the procedure has been included and can be found between lines 514 and 516 of the manuscript.

Reviewer 2 Report
Comments and Suggestions for Authors
The manuscript is well-structured and addresses a significant clinical need. However, major revisions are required to ensure the protocol's safety and scientific rigor: The guidance on propofol sedation is a critical concern. Stating that anaesthetist review is not routinely required for deep sedation is inconsistent with international safety standards and must be corrected. Additonally, The use of non-standard, subjective success criteria (e.g., the "King's triad," a 10% threshold for Partial Response) undermines the protocol's reproducibility. Adopting standardized, objective metrics (like RECIST criteria) is essential.
Author Response
We sincerely thank the reviewers for their thoughtful and constructive feedback. We appreciate the recognition of the clinical significance of this work and welcome the opportunity to enhance clarity, safety rationale, and reproducibility in the protocol. Below, we provide a detailed, point-by-point response to each comment, describing the revisions made to the manuscript.
Reviewer Comment:
- “The manuscript is well-structured and addresses a significant clinical need. However, major revisions are required to ensure the protocol's safety and scientific rigour. The guidance on propofol sedation is a critical concern. Stating that anaesthetist review is not routinely required for deep sedation is inconsistent with international safety standards and must be corrected.”
Response:
Thank you for highlighting the importance of sedation governance. We agree that clarity and alignment with international safety guidance are essential.
Revision Made:
We revised the sedation section to specify that Propofol sedation is administered only under anaesthetist-led deep sedation or general anaesthesia, consistent with international standards.
A new statement has been issued indicating that “the standard of care is propofol administered by Anaesthesiologists.” (lines 157 and 158 of the manuscript)
- “The use of non-standard, subjective success criteria (e.g., ‘King’s triad’, a 10% threshold for Partial Response) undermines reproducibility. Adopting standardized objective metrics (e.g., RECIST) is essential.”
Response:
We fully agree and thank the reviewer for this important point. The “King’s triad” consists of signs indicative of lesion ablation. It does not signify a response to treatment. This point was highlighted in the manuscript.
Revision Made:
The manuscript has been revised to include a description of the response to treatment based on standard terminology according to the RECIST criteria. This is highlighted in lines 667 to 679 of the manuscript. This revision enhances external reproducibility, comparability, and rigour.

Reviewer 3 Report
Comments and Suggestions for Authors
The review: “Establishing a Salvage Endoscopic Electroporation (SEE) Service for Colorectal Cancer: The King’s Protocol for Clinical Implementation” presents comprehensive knowledge on colorectal cancer salvage therapy.
Authors managed to make review of the literature and guidelines in the field. The objective of the study was to obtain the modern and comprehensive protocol for SEE in colorectal cancer.
In my opinion the aim of the study was achieved.
Author Response
Reviewer Comment:
“The review demonstrates extensive knowledge of colorectal cancer salvage therapy. The authors effectively evaluate the literature and guidelines in the field. The objective of the study was to develop a modern and comprehensive protocol for SEE in colorectal cancer. In my opinion, the aim of the study was achieved”.
Response:
We are grateful for the reviewer’s positive evaluation and are pleased that the manuscript’s objective and clinical relevance were clearly conveyed. We appreciate the acknowledgement that the protocol is comprehensive and grounded in current literature and practice standards.
